# A Study on Mortality Predictors in Hemodialysis Patients Infected with COVID-19: Impact of Vaccination Status

**DOI:** 10.3390/vaccines12010002

**Published:** 2023-12-19

**Authors:** Voin Brkovic, Gorana Nikolic, Marko Baralic, Milica Kravljaca, Marija Milinkovic, Jelena Pavlovic, Mirjana Lausevic, Milan Radovic

**Affiliations:** 1Clinic of Nephrology, University Clinical Centre of Serbia, 11000 Belgrade, Serbia; 2Faculty of Medicine, University of Belgrade, 11000 Belgrade, Serbia; 3Institute of Pathology, Faculty of Medicine, University of Belgrade, 11000 Belgrade, Serbia

**Keywords:** COVID-19, vaccines, chronic kidney disease, hemodialysis, lymphocytopenia

## Abstract

The global outbreak of COVID-19, caused by the severe acute respiratory syndrome coronavirus 2 (SARS-CoV-2), has prompted significant public health concerns. This study focuses on 442 chronic hemodialysis patients diagnosed with COVID-19, emphasizing the impact of vaccination status on clinical outcomes. The study investigates the correlation between vaccination status and laboratory findings, aiming to identify predictive factors for mortality. Results indicate that vaccination status plays a crucial role in outcomes. Full vaccination, evidenced by two or three doses, is associated with better outcomes, including reduced incidence of bilateral pneumonia and lower risks of complications such as hemorrhage and thrombosis. Laboratory analyses reveal significant differences between vaccinated and unvaccinated patients in parameters like C-reactive protein, ferritin, and white blood cell counts. Univariate and multivariate Cox proportional hazards regression analyses identify several factors influencing mortality, including comorbidities, pneumonia development, and various inflammatory markers. In conclusion among hemodialysis patients affected by COVID-19 infection, vaccination with at least three doses emerges as a protective factor against fatal outcomes. Independent predictors of mortality are CRP levels upon admission, maximum CRP values during the illness and cardiovascular comorbidities. Noteworthy lymphocytopenia during infection exhibits a notable level of specificity and sensitivity in predicting mortality.

## 1. Introduction

The global outbreak of coronavirus disease 2019 (COVID-19) signifies a widespread pandemic resulting from the infectious severe acute respiratory syndrome coronavirus 2 (SARS-CoV-2) [1,2,3]. This viral agent is responsible for causing the respiratory illness that has affected populations worldwide [4,5]. Because of the gravity of the situation, the World Health Organization declared COVID-19 a public health emergency of international concern in January 2020 [6], and announced the pandemic in March 2020 [7].

The incidence of COVID-19 is notably elevated among the elderly and those with comorbidities [8,9,10]. Pre-existing conditions such as cardiovascular disease, diabetes mellitus (DM), hypertension (HT), and chronic kidney disease (CKD) have been identified as significant risk factors for severe disease and mortality [11,12,13,14,15,16]. The emergence of the COVID-19 pandemic has brought forth unprecedented challenges for individuals on a global level, but for those undergoing hemodialysis, the impact has been particularly pronounced [17,18,19,20].

Considering that most individuals with chronic kidney disease (CKD) experience compromised immune systems due to uremia, requiring regular dialysis sessions at hemodialysis centers two to three times a week, this group is more vulnerable to SARS-CoV-2, exhibiting a higher mortality rate compared to the general population [14,21].

Additionally, various laboratory indicators, including lymphocytopenia, leukocytosis, elevated C-reactive protein (CRP), D-dimer, procalcitonin, and troponin I, have been linked to increased mortality, especially among hemodialyzed patients [22,23,24]. Moreover, a connection has been observed between the use of mechanical ventilation and admission to intensive care units (ICUs) with elevated mortality rates [25].

A crucial element in safeguarding CKD patients is implementing a successful vaccination strategy against SARS-CoV-2. This is especially vital due to the observed correlation between the severity of COVID-19 and the stage of CKD in these individuals. Those with kidney failure tend to exhibit the most severe symptoms of COVID-19 [26]. Interestingly, patients with kidney failure appear to mount a sufficient cellular and humoral response to a natural SARS-CoV-2 infection. The magnitude and functionality of SARS-CoV-2-specific immunity, including T cell and neutralizing antibody responses, were found to be comparable between these patients and individuals with normal kidney function [27,28]. In general, individuals with chronic kidney disease tend to have diminished immune responses to vaccinations when compared to those who are in good health [29].

The main goal of the research is to ascertain the optimal number of vaccine doses for dialysis patients that offer enhanced protection against lethal outcomes. Furthermore, the study aims to identify independent predictors of lethal outcomes in these patients, adjusted for vaccination status. Secondary objectives encompass evaluating the impact of the quantity of administered SARS-CoV-2 vaccine doses on averting the emergence of severe clinical manifestations of COVID-19 infection, as evidenced by the severity of respiratory failure and adverse vascular events.

## 2. Materials and Methods

This prospective observational cohort study included 442 patients on a chronic hemodialysis treatment program affected by COVID-19 infection, conducted at the COVID hospital in Batajnica University Clinical Center of Serbia, from December 2020 to June 2022. Consecutive inclusion in the study involved patients on a chronic hemodialysis treatment program with confirmed infection with the SARS-CoV-2 virus. The Ethic Committee of the Faculty of Medicine University of Belgrade, Serbia, granted approval to collect the medical data, and carry out the study (ref. number 17/I-10). The study presented here was conducted following all ethical standards laid down in the 1964 Declaration of Helsinki.

### 2.1. Study Protocol

The inclusion criteria encompassed patients aged 18 and older who had been on a chronic hemodialysis treatment program for more than 3 months and had a confirmed infection with the SARS-CoV-2 virus by PCR test. The exclusion criteria comprised patients younger than 18 years old and patients who were transiently dialyzed due to acute kidney injury. In the case of a confirmed infection with the SARS-CoV-2 virus, regardless of whether they have symptoms or not, patients were referred to the COVID hospital in Batajnica for further treatment. From the beginning of the epidemic and throughout the study duration, all facilities implemented a protocol involving temperature and symptom screening at each dialysis session. Additionally, all patients underwent SARS-CoV-2 PCR testing, with extra PCR testing conducted for individuals who had been in contact with confirmed cases. Cases were also identified through testing prompted by contact with a confirmed case or the presence of symptoms, such as seeking emergency services. By implementing this strategy from the very beginning of the epidemic in Serbia, the number of potential undetected asymptomatic individuals infected with SARS-CoV-2 was minimal. The date of SARS-CoV-2 infection was determined based on the date of the initial positive PCR result within the observation period.

The patient monitoring period in the study encompasses the time from the registration of the presence of the SARS-CoV-2 virus in patients with either (a) positive epidemiological risk, involving direct contact with a person infected with the SARS-CoV-2 virus, or (b) the onset of symptoms of COVID-19 infection, until the outcome.

The outcome was defined as (1) discharge from the COVID hospital in Batajnica in the event of the cessation of the need for oxygen therapy with a negative test for the presence of the SARS-CoV-2 virus, or (2) occurrence of a fatal outcome.

Demographic data, data on the length of dialysis vintage, and anamnestic data on the onset of COVID-19 symptoms were collected upon the patient’s admission to the COVID hospital in Batajnica, supplemented from accompanying medical documentation. Data on the course of the disease, laboratory analyses, chest X-ray findings, modes of oxygen supplementation, and administered drug therapy, as well as the treatment outcome, were extracted from the medical history. Information about the vaccination of patients against the SARS-CoV-2 virus were retrieved from the official database of the Ministry of Health of the Republic of Serbia, accessible through the internet portal. The schematic layout of study protocol is presented in Figure 1.

### 2.2. Measurement Instruments

The presence of pneumonia foci will be analyzed using tele-radiography of the lungs in the postero-anterior position. Based on the findings, patients were categorized into groups: (1) those without pneumonia, (2) those with unilateral pneumonia, and (3) those with bilateral pneumonia. Monitoring of inflammatory parameters and blood counts were conducted through standard laboratory analyses. The adequacy of pulmonary function will be assessed by arterial blood oxygen saturation using a gas analyzer (Siemens AG, Munich, Germany).

### 2.3. Statistical Analysis

The normal distribution of continuous variables was tested using the Kolmogorov–Smirnov test. Continuous variables with a normal distribution are presented as mean  ±  standard deviation (SD), and continuous variables that did not show a normal distribution are presented as the median value and interquartile range (IQR). They were compared using one-way ANOVA, or its non-parametric equivalent Kruskal–Wallis test. Leven’s test for homogeneity of variance was used to test for equality of variances. Tokey test was performed for post hock multiple comparisons or Mann–Whitney U test for non-normal distributed variables. Categorical variables are presented as counts and percentages, and were compared with the Chi square or Fisher’s exact test, as appropriate. Survival curves for different vaccination status groups of patients were constructed using the Kaplan–Meier method and further compared with the log-rank test. The identification of independent predictors for death was determined using a Cox proportional hazards regression model. Initially, all variables underwent univariate analysis. Those variables demonstrating an association with the outcome at a significance level below 0.1 were subsequently incorporated into the multivariate Cox model. The C statistic, a measure of the area under the receiver operating characteristic curve (AUC), quantified the predictive validity form mortality of CRP on admission, maximal value of CRP and minimal lymphocyte counts during hospitalization.

Statistical analyses were performed using the statistical package for social sciences, version 17 (SPSS, Chicago, IL, USA). Statistical significance was defined as *p* < 0.05.

## 3. Results

This research included 442 patients on maintenance hemodialysis, who received treatment at the COVID hospital in Batajnica, University Clinical Center of Serbia, between December 2020 and June 2022 due to SARS-CoV-2 infection. During that period, in accordance with global epidemiological data, the prevailing variants of the SARS-CoV-2 virus were primarily identified as the beta and delta variants [30]. Demographic characteristics and vaccination status details are presented in Table 1. Among the 442 patients, 173 were unvaccinated against COVID-19, 19 had received a single dose, 112 had received two doses, and 138 had received three doses. After analyzing gender, age, primary kidney disease diagnosis, and comorbidities, no statistically significant differences were identified.

The majority of patients (74%) were administered the inactivated vaccine (Sinopharm, Beijing, China), with the remaining individuals receiving mRNA vaccine (Pfizer, New York, NY, USA) at 8.6%, the vector vaccine (Sputnik V, Moscow, Russia) at 5.2%, and vector vaccine (AstraZeneca, Cambridge, UK) at 5.2%. Sinopharm was selected for the first two vaccine doses in 7.1% of patients, with Pfizer being chosen for the third dose, Figure 2.

All patients admitted to the hospital undergo a chest X-ray (Table 2). Among the 442 patients, 189 showed no signs of pneumonia, while 227 exhibited bilateral pneumonia. Notably, 107 of these were unvaccinated, and 47 received three vaccine doses (*p* < 0.001). Also, a difference was shown only among patients who received three doses of the vaccine in comparison to the others. A difference was shown between patients who received three doses, compared to the unvaccinated and those who received two doses, while no difference was observed among the other cohorts. Out of the total 442 patients, 154 required additional oxygen support, with 66 patients on invasive mechanical ventilation (IMV). Of those on IMV, 34 were unvaccinated, and eight received three vaccine doses (*p* < 0.001). On post hoc analysis, a difference was observed exclusively among patients who received three doses compared to the others. According to our data, the use of antiviral drugs and monoclonal antibody—RegenCov (casirivimab and imdevimab)—did not demonstrate statistical significance between unvaccinated and vaccinated patients. Upon hospital admission, there were significant differences in oxygen saturation between vaccinated and unvaccinated patients (*p* = 0.017). Notably, patients who received two or three doses of the vaccine demonstrated higher oxygen saturation compared to unvaccinated patients. Regarding the systolic blood pressure, on admission, no differences were observed between vaccinated and unvaccinated patients (Table 2).

In the absence of prior indications of bleeding, heparin was administered to all patients during hemodialysis. On non-dialysis days, low molecular weight heparin (LMWH) was prescribed, with continuous monitoring of anti-Xa levels to ensure they remained within the prophylactic range. During the hospital stay, complications such as hemorrhage and thrombosis exhibited significant differences between vaccinated and unvaccinated patients (*p* = 0.022), whereas 20 patients of total 34 with hemorrhage were unvaccinated and five of them received all three doses. Thrombosis gained 12 unvaccinated patients out of 21, and four were fully vaccinated patients. In post hoc analysis, it has been demonstrated that this distinction is associated with both the unvaccinated individuals and those who have received three doses (*p* = 0.020). No cases of Heparin-Induced Thrombocytopenia (HIT) were recorded.

A total of 119 patients experienced a fatal outcome, with 60 being unvaccinated and 17 having received the full vaccination series (*p* < 0.001). Post hoc analysis revealed that the distinction is applicable to patients who have received three doses, with no discernible difference observed between the unvaccinated individuals and those who received two doses (Table 2).

During the admission of patients to the hospital, biochemical analyzes were performed, with a specific emphasis on inflammatory markers (Table 3). A significant and statistically notable difference in C-reactive protein (CRP) levels in serum was observed between unvaccinated and vaccinated patients (*p* < 0.001). Significantly lower CRP levels were observed in individuals who received all three vaccine doses when compared to both unvaccinated individuals and those who received two doses.

A significant difference in the concentrations of serum ferritin was established among patients based on vaccination status (*p* < 0.001). Patients who received three or two doses exhibited significantly lower serum ferritin values compared to the unvaccinated ones. Furthermore, there were significant differences in blood albumin levels among the groups (*p* < 0.001).

Throughout the hospitalization period, patients underwent daily monitoring for biochemical analysis (Table 4). Significant differences were observed in the maximum levels of white blood cells (WBC) between unvaccinated and vaccinated individuals (*p* = 0.001). The detected lymphocytopenia, representing minimal lymphocyte values during hospitalization. It was least pronounced in patients who received three vaccine doses.

Significant differences in the maximum values of CRP were observed between unvaccinated and vaccinated patients (*p* < 0.001). Significantly lower values were observed in patients who were vaccinated with three doses compared to the others.

When comparing maximum ferritin levels between unvaccinated and vaccinated patients, a substantial difference was noted (*p* < 0.001). The median values in unvaccinated patients were 1976.2 µg/L (IQR 3853.70), whereas in patients who received all three doses of the vaccine, the median was 766.3 µg/L (IQR 1893.4) (*p* = 0.002), Table 4.

Furthermore, variations in the highest fibrinogen levels were noted between unvaccinated and vaccinated individuals (*p* = 0.001). Unvaccinated patients exhibited a maximal mean fibrinogen value of 5.7 ± 1.89 g/L, whereas fully vaccinated patients showed a lower maximal mean fibrinogen value of 5.1 ± 1.69 g/L, as detailed in Table 4.

Through Univariate Cox proportional hazards regression analysis, our findings indicate that elderly patients have an increased risk of a lethal outcome (HR: 1.025, 95% CI 1.010–1.040). While patients with comorbidities generally did not exhibit a heightened risk, those with CMP (HR: 2.400, 95% CI 1.666–3.458) showed a statistically significant difference (*p* < 0.001). Also, only bilateral pneumonia was a predictor of a lethal outcome in the examined patients (HR: 19.888, 95% CI 6.312–62.664). Mortality was nearly 2.5 times higher in patients who had complications in the form of bleeding compared to patients without vascular complications and those who developed thrombosis (HR 2.401, 95% CI 1.539–3.745; Table 5).

In the Multivariate Cox proportional hazards regression analysis, our findings indicate a significant increase in the mortality of patients with CM (HR: 3.364, 95% CI 1.874–6.040). Additionally, CRP on admission (HR: 1.004, 95% CI 1.001–1.008), maximal values of CRP (HR: 1.006, 95% CI 1.002–1.010), and maximal values of IL-6 (HR: 1.001, 95% CI 1.0001–1.002) emerged as independent risk factors for a lethal outcome, as presented in Table 6.

Additionally, Univariate Cox proportional hazards regression analysis demonstrated that only patients vaccinated with three doses had significantly lower mortality compared to the unvaccinated (HR 0.397, 95% CI 0.231–0.683).

Various inflammatory parameters, including WBC on admission (HR: 1.077, 95% CI 1.038–1.118), lymphocyte count on admission (HR: 0.425, 95% CI 0.274–0.660), CRP on admission, and maximal values during hospitalization (HR: 1.006, 95% CI 1.004–1.007, HR: 1.007, 95% CI 1.005–1.009), IL-6 on admission and maximal values during hospitalization (HR: 1.001, 95% CI 1.000–1.002, HR: 1.001, 95% CI 1.000–1.002), ferritin on admission and maximal values during hospitalization (HR: 1.001, 95% CI 1.000–1.002, HR: 1.001, 95% CI 1.001–1.002), D-dimer on admission and maximal values during hospitalization (HR: 1.054, 95% CI 1.033–1.076, HR: 1.022, 95% CI 1.012–1.033), as well as minimal and maximal counts of lymphocytes (HR: 0.062, 95% CI 0.027–0.143, HR: 0.690, 95% CI 0.500–0.952), exhibited significant influence as predictive factors for a lethal outcome (Table 5).

In Figure 3, the Kaplan–Meier survival analysis curve is presented according to the number of received vaccine doses. Survival was significantly higher only in patients vaccinated with three doses (*p* = 0.007).

The significance of lymphocytopenia in predicting a lethal outcome has been established in dialysis patients, both across the entire examined population and within patient subgroups based on their vaccination status. A cutoff of 0.395 for the minimal amount of lymphocytes during hospitalization shows a sensitivity of 84.2% and specificity of 81.5% for mortality prediction in the study population, Figure 4.

The C-statistic revealed a high level of discriminatory ability for the maximum CRP values in predicting mortality among dialysis patients, thought all subgroups based on the number of received doses of the vaccines (Figure 5). The cut-off value for CRP upon admission set at 56.5 mg/dL, demonstrates a sensitivity of 69.7% and specificity of 67.5% for mortality prediction. On the other hand, the cut-off value for the maximal CRP during hospitalization, set at 108.15 mg/dL, exhibits a sensitivity of 82.4% and specificity of 76.2% for mortality prediction.

## 4. Discussion

Patients with chronic kidney disease (CKD) are a particularly vulnerable group to infectious diseases, and morbidity and mortality increase with the progression of CKD [31,32]. Considering that patients undergoing chronic hemodialysis treatment often have one or more comorbidities, this makes them an even more vulnerable group. One of the reasons for such outcomes is the immunocompromised state of these patients. On the other hand, immunodeficiency leads to a weaker immune response to active immunization with vaccines. In an extensive meta-analysis, Notarte et al. demonstrated that individuals undergoing maintenance hemodialysis consistently exhibited lower antibody levels across all post-vaccination periods compared to the general population. Among the cohorts studied, individuals with chronic kidney disease displayed the most robust antibody immune response, followed by those participating in hemodialysis, while kidney transplant recipients exhibited the lowest response following SARS-CoV-2 vaccination. [33].

Our findings indicate that factors such as gender, primary kidney disease diagnosis, and comorbidities did not exhibit statistical significance in hemodialyzed patients when considering vaccination status and mortality, except for age and cardiomyopathy. In the study by Rista et al., higher mortality rates were observed in patients undergoing dialysis for diabetic nephropathy (*p* < 0.04) and peripheral vascular disease (*p* < 0.01) [34], in contrast to Selvaskandan et al., who found no association between obesity, diabetes status, ethnicity, or Charlson Comorbidity Index with COVID-19 severity [35]. Kikuchi et al., in their study, used multivariate analysis to show a significant increase in mortality with prolonged duration of dialysis, without a difference in gender or primary disease [36]. Some multicenter studies indicated that advanced age, diabetes, and immune suppression in patients on hemodialysis with SARS-CoV-2 infection correlated with more severe illness [37], and in another study, age and pneumonia were independent risk factors for death using multivariable analysis [38]. The presence of multiple comorbidities and the prolonged duration of chronic kidney disease until the terminal stage constitute risk factors for the development of diseases in various organs and organ systems, particularly affecting the cardiovascular system. Patients in this category often experience an overall compromised state. In such a context, the frailty clinical index emerges as a noteworthy and comprehensive indicator of the general condition of dialysis patients. Numerous studies have underscored its significance as a predictive factor for adverse and potentially lethal outcomes [39,40]. Some of these differences can be explained by the inherent heterogeneity in the design of studies encompassing these patients. Certainly, a common observation was that dialysis patients are at a higher risk of a fatal outcome from a COVID-19 infection compared to the general population.

Laboratory findings played a significant role in determining the survival rate. Our results indicated that various inflammatory parameters, including WBC on admission lymphocyte count on admission CRP on admission, and maximal values during hospitalization IL-6 on admission and maximal values during hospitalization, ferritin on admission and maximal values during hospitalization, D-dimer on admission and maximal values during hospitalization as well as minimal and maximal counts of lymphocytes, significantly influenced the predictive factors for a lethal outcome.

Zaidi et al., in their meta-analysis, suggested that lymphocytopenia serves as a crucial hematological indicator of severe COVID-19, with lymphocytopenia < 1500/mm^3^ being a practical parameter for predicting severe outcomes [41]. Niu et al. demonstrated the predictive efficacy of lymphocytopenia, with areas under the receiver operating characteristic curves (AUC) of 0.68, 0.69, 0.78, and 0.79 for the corresponding adverse outcomes when age, gender, race, and comorbidities were taken into account. Among the patients analyzed, 2409 (57.3%) exhibited lymphocytopenia (absolute lymphocyte count < 1.1 × 109/L) upon admission, experiencing higher rates of ICU admission (17.9% versus 9.5%, *p* < 0.001), invasive mechanical ventilation (14.4% versus 6.5%, *p* < 0.001), dialysis (3.4% versus 1.8%, *p* < 0.001), and in-hospital mortality (16.6% versus 6.6%, *p* < 0.001). Multivariable-adjusted odds ratios for these outcomes were 1.86 (95% CI, 1.55–2.25), 2.09 (95% CI, 1.69–2.59), 1.77 (95% CI, 1.19–2.68), and 2.19 (95% CI 1.76–2.72), respectively, compared to those without lymphocytopenia. These findings were not compared with vaccination status [42]. Hartantri et al., based on laboratory parameters, identified high procalcitonin and C-reactive protein levels, along with a neutrophil-to-lymphocyte ratio >3.53, as associated with mortality. In our study, CRP on admission as well as maximal values during hospitalization were found to be independent predictive factors for a lethal outcome, respectively [43]. Rista et al., using ROC analysis, identified lymphocytopenia and eosinopenia as robust predictors of mortality. Post-vaccination, the mortality rate in the vaccinated population was 8%, a significant contrast to the 66.7% mortality rate observed in the unvaccinated group (*p* < 0.001) [34]. Similar results were observed in our study. We also found a significant predictive ability of lymphocytopenia for mortality in dialysis patients. Additionally, the AUC was 0.886 and a cutoff of 0.395 for the minimal amount of lymphocytes during hospitalization shows a sensitivity of 84.2% and specificity of 81.5% for mortality prediction in the study population. The C-statistic indicated a robust discriminatory ability in predicting mortality among dialysis patients using the maximum CRP values, across all subgroups based on the number of vaccine doses received. A threshold value for CRP upon admission, established at 56.5 mg/dL, shows a sensitivity of 69.7% and specificity of 67.5% for mortality prediction. Conversely, the threshold value for maximum CRP during hospitalization, set at 108.15 mg/dL, demonstrates a sensitivity of 82.4% and specificity of 76.2% for predicting mortality.

Upon admission to the hospital, only patients with the most severe conditions were prescribed antiviral medication based on its availability, given the shortage of medications for COVID-19 during that period. Although many studies showed the influence of the usage of antiviral drugs in contrary to the duration of hospitalization [36,44], our results showed that antiviral drugs and monoclonal antibodies did not show any statistical significance in mortality prevention nor duration of hospitalization (*p* = 0.196). A possible explanation lies in the fact that these medications were administered only to the most severe patients and, therefore, there is a selection bias.

Numerous studies have highlighted the impact of vaccination status on survival rates in hemodialyzed patients with COVID-19, presenting real-world effectiveness and immunogenicity data for vaccines [45]. Espi et al. conducted a prospective observational study, emphasizing the safety and efficacy of a third dose of mRNA vaccine in maintenance hemodialysis patients [46]. In a study by Ashby et al., the analysis demonstrated that, factoring in age, comorbidities, and the time period, a two-dose vaccination history was associated with a 75% lower risk of admission and an 88% reduction in deaths compared to unvaccinated patients. Even after the first dose, there were noticeable differences, with a 45% lower risk of admission. Notably, the protection from severe illness associated with vaccination was most evident in patients over 65 years. Third doses provided additional protection, with a 51% further reduction in admissions [47]. Furthermore, a US study involving over 12,000 hemodialysis patients receiving vaccine indicated a significantly lower subsequent risk of symptomatic COVID-19 compared to a matched unvaccinated cohort dialyzing at the same facilities [45]. However, an analysis of linked Scottish registry data, encompassing patients on dialysis and those with kidney transplants, estimated vaccine effectiveness in preventing infection at only 33% [48]. The results of our study indicated that patients who received three doses of the vaccine against SARS-CoV-2 had a 60.3% lower probability for lethal outcome in comparison to unvaccinated patients. Additionally, patients vaccinated with less than three doses did not show a significantly lower mortality compared to the unvaccinated group.

In our results, a significant difference was observed in the presence of bilateral pneumonia according to vaccination status. The lowest incidence of bilateral pneumonia was observed in patients who received a three-dose vaccination. However, univariate Cox analysis showed a higher risk for a lethal outcome of bilateral pneumonia (HR: 19.888). Upon hospital admission, there were notable differences in oxygen saturation between vaccinated and unvaccinated patients (*p* = 0.017). Among the total 442 patients, 154 required additional oxygen support, with 66 undergoing invasive mechanical ventilation (IMV). Among those on IMV, 34 were unvaccinated, and eight had received three vaccine doses (*p* < 0.001). This finding supports the fact that severe impairment of respiratory function has a poor prognostic significance, especially in such a vulnerable category of patients [38].

To the best of our knowledge, there is no literature data regarding the influence of vaccination status (unvaccinated/vaccinated) on complications such as hemorrhage in hemodialyzed patients with COVID-19 infection, particularly. Giot et al. demonstrated only the safety of vaccine application against COVID-19 during the dialysis session in patients under oral anticoagulants therapy [49]. Our results revealed a significantly higher incidence of hemorrhage as a complication due to COVID-19 infection within unvaccinated patients than in fully vaccinated patients (*p* < 0.001, HR: 2.401, 95%CI 1.539–3.745), which therefore represents a potential predictive factor for a lethal outcome.

## 5. Conclusions

Among hemodialysis patients affected by COVID-19 infection, vaccination with at least three doses emerges as a protective factor against fatal outcomes and bilateral pneumonia. Predictors of mortality in this cohort include CRP levels upon admission, especially the maximum CRP values during the illness. Additionally, individuals with cardiovascular comorbidities face a heightened risk of adverse consequences. Noteworthy is the observation that the most pronounced lymphocytopenia during infection exhibits a notable level of specificity and sensitivity (84.2% and 81.5%, respectively) in predicting mortality, with a lymphocyte count cutoff of 0.395 × 10^9^/L. Our findings suggest that hemodialysis patients constitute a particularly vulnerable group, necessitating a significantly more robust vaccination approach to prevent severe clinical events, including fatal outcomes.

## Figures and Tables

**Figure 1 vaccines-12-00002-f001:**
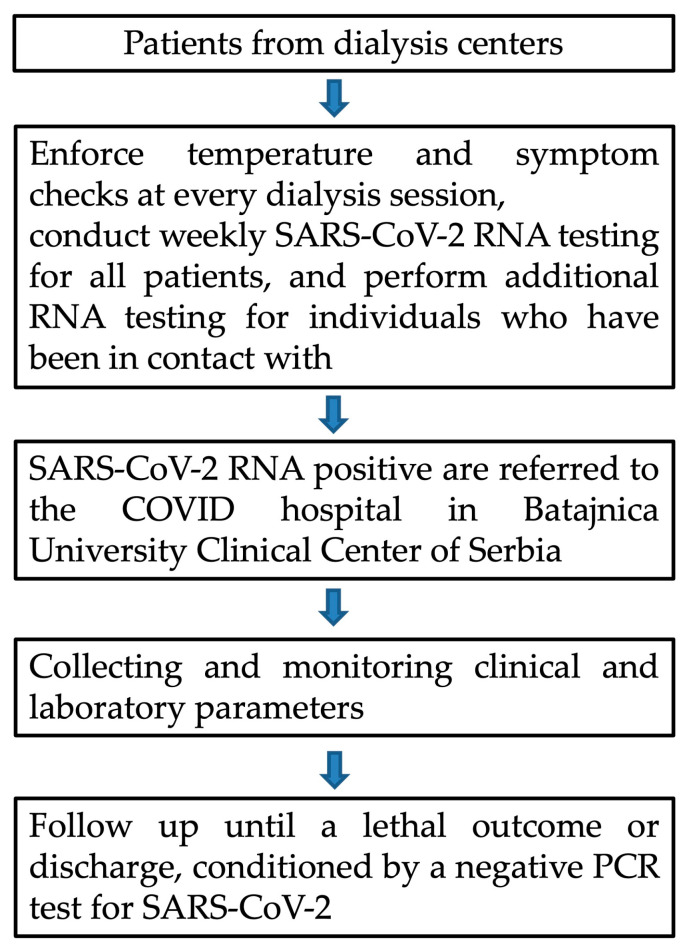
Schematic layout of study protocol, point by point.

**Figure 2 vaccines-12-00002-f002:**
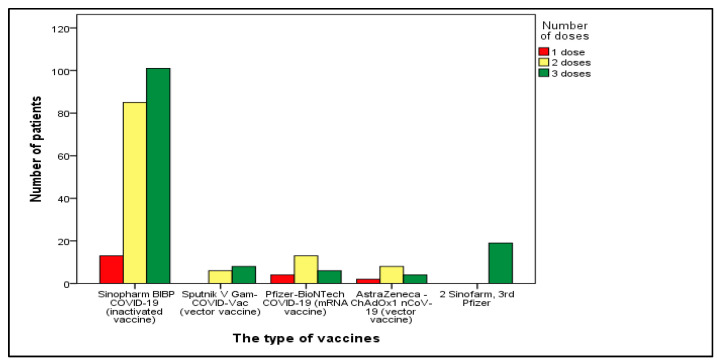
Distribution of patients based on the type and number of received vaccines.

**Figure 3 vaccines-12-00002-f003:**
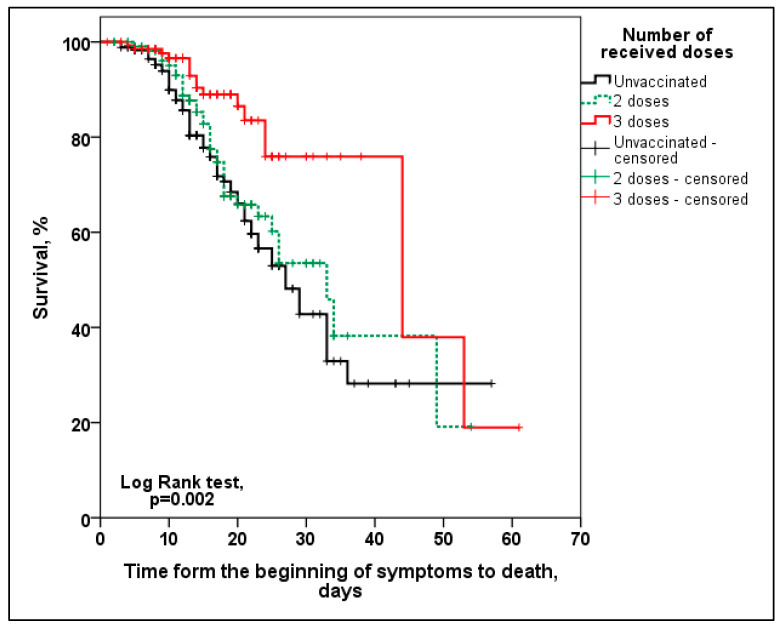
Kaplan–Meier curve survival analysis according to the number of received vaccine doses whereas unvaccinated vs. 3 doses *p* = 0.001, 2 doses vs. 3 doses *p* = 0.009.

**Figure 4 vaccines-12-00002-f004:**
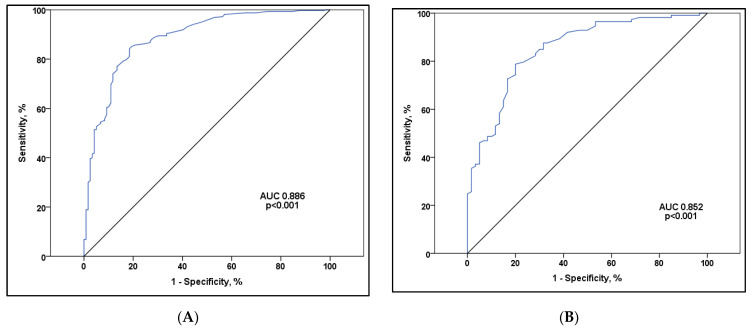
The ROC curve of the discriminative ability of the minimum number of lymphocytes during hospitalization in predicting mortality: (**A**) represents all admission patients to the hospital (n = 442), (**B**) represents unvaccinated patients (n = 173), (**C**) represents patients who received two doses of the vaccine (n = 112) and (**D**) represents fully vaccinated patients with three doses of the vaccine (n = 138).

**Figure 5 vaccines-12-00002-f005:**
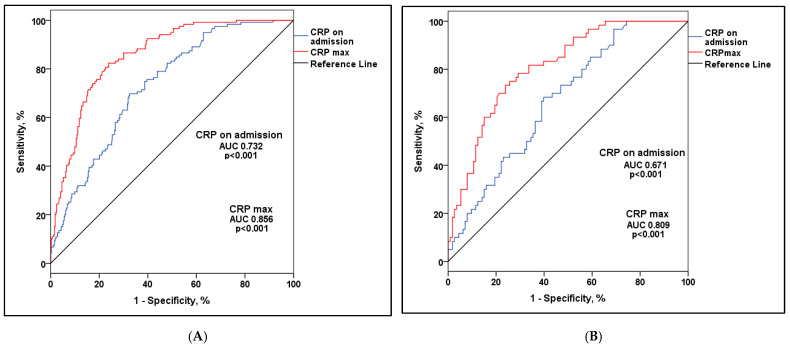
ROC curve of the discriminative ability of CRP values in predicting mortality: (**A**) represents all admission patients to the hospital (n = 442), (**B**) represents unvaccinated patients (n = 173), (**C**) represents patients who received two doses of the vaccine (n = 112) and (**D**) represents fully vaccinated patients with three doses of the vaccine (n = 138).

**Table 1 vaccines-12-00002-t001:** Demographic and clinical characteristics of the studied population according to vaccination status.

	Overall, N = 442	Unvaccinated, n = 173	1 dose, n = 19	2 doses, n = 112	3 doses, n = 138	*p*
Male sex, n (%)	285 (64.5%)	118 (68.2)	9 (47.7)	68 (60.7)	90 (65.2)	0.240
Age, years ± SD	63 ± 13.7	62 ± 14.3	63 ± 15.9	64 ± 13.9	65 ± 12.3	0.162
Primary kidney disease, n (%)						0.502
Hypertension	169 (38.2)	62 (35.8)	9 (47.4)	43 (38.4)	55 (39.9)	
Diabetes	115 (26.0)	42 (24.3)	3 (15.8)	33 (29.5)	37 (26.8)	
Glomerulonephritis	36 (8.1)	17 (9.8)	2 (10.5)	5 (4.5)	12 (8.7)	
Lupus nephritis and vasculitis	21 (4.8)	9 (5.2)	3 (15.8)	4 (3.6)	5 (3.6)	
Polycystic kidney disease	27 (6.1)	9 (5.2)	1 (5.3)	7 (6.3)	10 (7.2)	
Others	74 (16.7)	34 (19.7)	1 (5.3)	20 (17.9)	19 (13.8)	
Comorbidity, n (%)						
Hypertension	374 (84.6)	144 (83.2)	16 (84.2)	94 (83.9)	120 (87.0)	0.831
Diabetes	125 (28.3)	47 (27.2)	3 (15.8)	35 (31.3)	40 (29.0)	0.554
Cardiomyopathy	155 (35.1)	64 (37.0)	9 (47.4)	32 (28.6)	50 (36.2)	0.295
Cerebrovascular disease	32 (7.2)	14 (8.1)	2 (10.5)	9 (8.0)	7 (5.1)	0.668
Malignancy	45 (10.2)	15 (8.7)	2 (10.5)	12 (10.7)	16 (11.6)	0.857
COPD	17 (3.8)	7 (4.0)	0 (0)	5 (4.5)	5 (3.6)	0.822
Duration of dialysis, years, median (IQR)	4 (7)	4.73 (8)	3.75 (9)	4 (7)	4 (7)	0.920

SD—standard deviation, IQR—interquartile range, COPD—chronic obstructive pulmonary disease.

**Table 2 vaccines-12-00002-t002:** Clinical characteristics and therapeutic approaches of patients infected with the SARS-CoV-2 virus based on vaccination status.

	Overall, N = 442	Unvaccinated, n = 173	1 dose, n = 19	2 doses, n = 112	3 doses, n = 138	*p*
The X-ray signs of pneumonia, n (%)						
Without pneumonia	189 (42.8)	50 (28.9)	5 (26.3)	46 (41.1)	88 (63.8)	<0.001
Unilateral pneumonia	26 (5.9)	16 (9.2)	2 (10.5)	5 (4.5)	3 (2.2)	
Bilateral pneumonia	227 (51.3)	107 (61.8)	12 (63.2)	61 (54.5)	47 (34.1)	
Oxigenation, n (%)						
Ambient air	268 (60.6)	83 (48.0)	10 (52.6)	64 (57.1)	111 (80.4)	<0.001
Low flow nasal cannula	83 (18.8)	40 (23.1)	7 (36.8)	22 (19.6)	14 (10.1)	
High Flow face mask	10 (2.3)	5 (2.9)	0 (0)	3 (2.7)	2 (1.4)	
Noninvasive ventilation	15 (3.4)	11 (6.4)	0 (0)	1 (0.9)	3 (2.2)	
Invasive mechanical ventilation	66 (14.9)	34 (19.7)	2 (10.5)	22 (19.6)	8 (5.8)	
Antivirotic, n (%)	59 (13.3)	19 (11.0)	1 (5.3)	14 (12.5)	25 (18.1)	0.196
Regen Cov, n (%)	22 (5.0)	6 (3.5)	1 (5.3)	6 (5.4)	9 (6.5)	0.667
Time from the last vaccine dose to the onset of illness, day, median (IQR)	141 (170)	-	14 (225)	168 (120)	122.5 (171)	<0.001
Time from illness onset to outcome, day, median (IQR)	15 (11) min-max 1–61	15 (12)	15 (18)	17 (10)	14 (11)	0.137
Oxygen saturation on admission, % ± SD	96 ± 5.3	95 ± 6.6	95 ± 4.1	96 ± 4.9	96 ± 4.1	0.017
Systolic blood pressure on admission, mm/Hg, mean ± SD	130 ± 22.9	134 ± 25.4	130 ± 24.3	133 ± 17.7	130 ± 23.4	0.401
Complication, n (%)						
Without complication	383 (86.7)	140 (80.9)	16 (84.2)	98 (87.5)	129 (93.5)	0.022
Major bleeding	34 (7.7)	20 (11.6)	3 (15.8)	6 (5.4)	5 (3.6)	
Thrombosis	21 (4.8)	12 (6.9)	0 (0)	5 (4.5)	4 (2.9)	
Bleeding + thrombosis	4 (0.9)	1 (0.6)	0 (0)	3 (2.7)	0 (0)	
Death, n (%)	119 (26.9)	60 (34.7)	7 (36.8)	35 (31.3)	17 (12.3)	<0.001

SD—standard deviation, IQR—interquartile range.

**Table 3 vaccines-12-00002-t003:** Inflammatory parameters on admission according to vaccination status.

	Overall, n = 442	Unvaccinated, n = 173	1 dose, n = 19	2 doses, n = 112	3 doses, n = 138	*p*
WBC count *, mean ± SD	7.1 ± 3.77	7.3 ± 4.39	5.8 ± 1.82	7.1 ± 3.51	7.0 ± 3.30	0.371
Ly count *, median (IQR)	0.93 (0.70)	0.84 (0.67)	0.95 (0.54)	0.91 (0.73)	1.0 (0.66)	0.207
Hgb, g/L, mean ± SD	104 ± 17.3	103 ± 19.6	99 ± 16.4	106 ± 16.0	104 ± 15.0	0.213
D-dimer, mg/L, median (IQR)	1.4 (1.72)	1.89 (2.30)	1.64 (1.59)	1.30 (1.77)	1.22 (1.05)	0.073
CRP, mg/L, median (IQR)	43.1 (82.98)	57.1 (98.05)	45.9 (85.5)	45.5 (72.5)	25.9 (63.45)	<0.001
Ferritin, µg/L, median (IQR)	923.3 (1806.48)	1108.0 (1801.80)	1557.3 (1748.3)	753.0 (1678.1)	624.8 (1287.0)	<0.001
Fibrinogen, g/L, mean ± SD	4.9 ± 1.61	5.1 ± 1.76	5.5 ± 1.52	5.1 ± 1.5	4.8 ± 1.48	0.127
Albumin, g/L, mean ± SD	36 ± 5.8	34 ± 6.0 *	36 ± 6.7	36 ± 6.0	37 ± 4.6	<0.001

* WBC—white blood cells, Ly—Lymphocyte, * multiply by 1 × 10^9^/L, CRP—C reactive protein, SD—standard deviation, IQR—interquartile range Post hoc analysis: CRP: unvaccinated vs. 3 doses *p* < 0.001, 2 doses vs. 3 doses *p* = 0.001; Feritin: unvaccinated vs. 2 doses *p* < 0.001, unvaccinated vs. 3 doses *p* < 0.001, Albumini: unvaccinated vs. 2 doses *p* = 0.046, unvaccinated vs. 3 doses *p* < 0.001.

**Table 4 vaccines-12-00002-t004:** Maximum values of inflammatory factors depending on vaccination status.

	Overall, N = 442	Unvaccinated, n = 173	1 dose, n = 19	2 doses, n = 112	3 doses, n = 138	*p*
WBC max * count, mean ± SD	12.4 ± 8.05	13.5 ± 7.95	12.8 ± 6.36	13.4 ± 9.06	10.2 ± 7.06	0.001
Ly min ** counts, median (IQR)	0.6 (0.69)	0.49 (0.65)	0.47 (0.59)	0.56 (0.67)	0.80 (0.62)	<0.001
Ly max counts, median (IQR)	1.25 (4.47)	1.27 (1.05)	1.18 (0.71)	1.20 (0.83)	1.25 (0.80)	0.701
D-dimer max, mg/L, median (IQR)	2.16 (3.88)	2.9 (4.81)	2.19 (1.98)	2.20 (3.80)	1.42 (2.54)	0.002
CRP max, mg/L, median (IQR)	78.6 (153.3)	109.3 (149.30)	100.3 (153.3)	77.85 (181.88)	37.8 (94.78)	<0.001
Ferritin max, µg/L, median (IQR)	1277.8 (3322.1)	1976.2 (3853.70)	2309.8 (3011.8)	1104.4 (3448.8)	766.3 (1893.4)	<0.001
Fibrinogen max, g/L, mean ± SD	5.6 ± 1.82	5.7 ± 1.89	6.4 ± 1.69	5.8 ± 1.78	5.1 ± 1.69	0.001
IL6 max, pg/mL, median (IQR)	49.1 (164.25)	49.1 (231.25)	51.75 (71.82)	64.9 (186.84)	36.2 (190.61)	0.808

WBC white blood cells, Ly—Lymphocyte counts, * max—maximum values during hospitalization, ** minimal values during hospitalization, CRP—C reactive protein, SD—standard deviation, IQR—interquartile range. Post hoc analysis: Le max.: unvaccinated. vs. 3 doses *p* = 0.002, 2 doses vs. 3 doses *p* = 0.008. Ly min.: unvaccinated vs. 3 doses *p* < 0.001, 2 doses vs. 3 doses *p* = 0.001. D-dimer max.: unvaccinated vs. 3 doses *p* < 0.001, 2 doses vs. 3 doses *p* = 0.019. CRP max.: unvaccinated vs. 3 doses *p* < 0.001, 2 doses vs. 3 doses *p* < 0.001. Ferritin max.: unvaccinated vs. 2 doses *p* = 0.004, 0 doses vs. 3 doses *p* < 0.001, Fibrinogen max.: unvaccinated vs. 3 doses *p* = 0.016, 2 doses vs. 3 doses *p* = 0.010.

**Table 5 vaccines-12-00002-t005:** Univariate Cox proportional hazards regression analysis of predictors of lethal outcome.

	B	*p*	HR	CI 95%
Age	0.025	0.001	1.025	1.010–1.040
Male	−0.047	0.808	0.954	0.653–1.394
Hypertension	0.303	0.288	1.354	0.774–2.371
Diabetes	0.202	0.300	1.224	0.835–1.794
Cardiomyopathy	0.876	<0.001	2.400	1.666–3.458
Cerebrovascular disease	0.471	0.115	1.604	0.892–2.874
Malignancy	−0.044	0.877	0.957	0.546–1.676
COPD	0.021	0.968	1.021	0.375–2.776
The X-ray signs of pneumonia, n (%)				
Without pneumonia	-	-	-	-
Unilateral pneumonia	0.913	0.191	3.299	0.551–19.761
Bilateral pneumonia	0.586	<0.001	19.888	6.312–62.664
Vaccination:				
Unvaccinated	-	-	-	-
1 dose	−0.216	0.603	0.806	0.357–1.817
2 doses	−0.180	0.399	0.836	0.550–1.268
3 doses	−0.923	0.001	0.397	0.231–0.683
Antivirotic	−0.365	0.253	0.694	0.372–1.298
RegenCov	−0.805	0.176	0.447	0.139–1.435
Duration of dialysis	0.015	0.323	1.015	0.968–1.044
Complication				
Without complication	-	-	-	-
Hemorrhage	0.876	<0.001	2.401	1.539–3.745
Thrombosis	0.515	0.108	1.674	0.893–3.140
Hemorrhage + thrombosis	0.392	0.508	1.480	0.463–4.729
WBC od admission	0.075	<0.001	1.077	1.038–1.118
Ly on admission	−0.856	<0.001	0.425	0.274–0.660
CRP on admission	0.006	<0.001	1.006	1.004–1.007
IL-6 on admission	0.001	0.024	1.001	1.000–1.002
Ferritin on admission	0.001	<0.001	1.001	1.001–1.002
Fibrinogen on admission	0.063	0.279	1.065	0.950–1.193
D dimer on admission	0.053	<0.001	1.054	10.33–1.076
Le max	0.053	<0.001	1.055	1.038–1.071
Ly min	−2.781	<0.001	0.062	0.027–0.143
Ly max	−0.371	0.024	0.690	0.500–0.952
D-dimer max	0.022	<0.001	1.022	1.012–1.033
CRP max	0.007	<0.001	1.007	1.005–1.009
Ferritin max	0.001	<0.001	1.001	1.001–1.002
IL6 max	0.001	<0.001	1.001	1.001–1.002

COPD—chronic obstructive pulmonary disease, WBC—white blood cells, Ly—Lymphocyte counts, CRP—C reactive protein, max—maximum values during hospitalization, min—minimal values during hospitalization.

**Table 6 vaccines-12-00002-t006:** Multivariate Cox proportional hazards regression analysis of predictors of lethal outcome.

	B	*p*	HR	CI 95%
Cardiomyopathy	1.213	<0.001	3.364	1.874–6.040
CRP on admission	0.004	0.005	1.004	1.001–1.008
CRP max	0.006	0.003	1.006	1.002–1.010
IL6 max	0.001	<0.001	1.001	1.001–1.002

CRP—C reactive protein, max—maximum values during hospitalization.

## Data Availability

The data presented in this study are available on request from the corresponding author. The data are not publicly available due to privacy of patients.

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
