# Peer review of "A Study on Mortality Predictors in Hemodialysis Patients Infected with COVID-19: Impact of Vaccination Status"

_vaccines, 2023, doi:10.3390/vaccines12010002_

Round 1

Reviewer 1 Report

Comments and Suggestions for Authors

The paper, "A Study on Mortality Predictors in Hemodialysis Patients Infected with COVID-19: Impact of Vaccination Status," presented at the review focuses the authors' aim on the possibility that anti-Covid -19 vaccination may reduce the impact on mortality in dialysis patients infected with Covid-19, compared with the unvaccinated. This is a single-center, observational study, and it isn't clearly defined whether prospective or retrospective.

The study is numerically consistent, although it is spread over a time frame of 18 months during which SARS-CoV-2 infection presented differences due to virus mutations, the characterization of which is not provided in the text. Results are presented relative to unvaccinated and vaccinated patients with 1, 2, and 3 doses.

Lacking in the text are references to the type of vaccines used while the intervalence of time between vaccination and infectiousness onset is reported.

From this it is evident that the group of patients with a vaccination (moreover, very small and not comparable with other groups) had definitely not yet developed an initial immune response to the vaccine.

The paper is sufficiently well documented and provides an interesting amount of data even though the groups of vaccinees are not homogeneous numerically, and in particular the group of those who had received only 1 dose, does not deserve a stand-alone evaluation, especially in the survival curves.

So too, ROC curve results could be limited to the unvaccinated and fully vaccinated.

The discussion is somewhat verbose and repetitive of the results.

There are no concrete references on the use of antivirals and timing of administration.

Regarding the thrombosis-haemorrhage issue, there is no indication of the type of anticoagulant used in dialysis nor whether there were cases of HIT in the patients.

The presentation of data is not always straightforward by mixing clinical and laboratory related variable results.

Some acronyms are not adequately explained in the text.

Author Response

Dear Sirs/Madam,

Thank you for your kind suggestions to improve our manuscript. Please find attached the document containing all the corrections that we have made in our text.

Reviewer 1:

The paper, "A Study on Mortality Predictors in Hemodialysis Patients Infected with COVID-19: Impact of Vaccination Status," presented at the review focuses the authors' aim on the possibility that anti-Covid -19 vaccination may reduce the impact on mortality in dialysis patients infected with Covid-19, compared with the unvaccinated. This is a single-center, observational study, and it isn't clearly defined whether prospective or retrospective.

  1. This is a single-center, observational study, and it isn't clearly defined whether prospective or retrospective.

Response: Thank you for your observation and suggestion. We rephrased first sentence in Material and Method section adding the specific type of study: ”This prospective observational cohort study included 442 patients on a chronic hemodialysis treatment program affected by COVID-19 infection, conducted at the COVID hospital in Batajnica University Clinical Center of Serbia, from December 2020 to June 2022.

  1. The study is numerically consistent, although it is spread over a time frame of 18 months during which SARS-CoV-2 infection presented differences due to virus mutations, the characterization of which is not provided in the text. Results are presented relative to unvaccinated and vaccinated patients with 1, 2, and 3 doses.’’

Response: Unfortunately, during that period, there was no possibility of precisely determining which variant of the SARS-CoV-2 virus each individual patient was infected with. In the Result section we added this sentence: During that period, in accordance with global epidemiological data, the prevailing variants of the SARS-CoV-2 virus were primarily identified as the beta and delta variants.’’

  1. “Lacking in the text are references to the type of vaccines used while the intervalence of time between vaccination and infectiousness onset is reported.”

Response: We insert one more picture in the Result with the references to the type of the vaccines and their distribution with explanation: The majority of patients (74%) were administered the inactivated vaccine (Sinopharm), with the remaining individuals receiving mRNA vaccine (Pfizer) at 8.6%, the vector vaccine (Sputnik V) at 5.2%, and vector vaccine (AstraZeneca) at 5.2%. Sinopharm was selected for the first two vaccine doses in 7.1% of patients, with Pfizer being chosen for the third dose, Figure 2.’’

  1. “From this it is evident that the group of patients with a vaccination (moreover, very small and not comparable with other groups) had definitely not yet developed an initial immune response to the vaccine.”

Response: We agree with your comment.

  1. “The paper is sufficiently well documented and provides an interesting amount of data even though the groups of vaccines are not homogeneous numerically, and in particular the group of those who had received only 1 dose, does not deserve a stand-alone evaluation, especially in the survival curves.”

Response: Thank you for your helpful suggestion. Paragraph regarding the group of patients who received a single vaccine dose have been removed from the text and from survival curves.

  1. “So too, ROC curve results could be limited to the unvaccinated and fully vaccinated.”

Response: In line with the previous answer, we will omit the ROC curve that illustrates patients who have received only a single vaccine dose. Further in the text, corrected figures and figure captions are presented.

Figure 4: The ROC curve of the discriminative ability of the minimum number of lymphocytes during hospitalization in predicting mortality: A- represent all admission patients to the hospital (n=442), B- represent unvaccinated patients (n=173), C- represent patients who received two doses of the vaccine (n=112) and D- represent fully vaccinated patients with three doses of the vaccine (n=138).

Figure 5. ROC curve of the discriminative ability of CRP values in predicting mortality: A- represent all admission patients to the hospital (n=442), B- represent unvaccinated patients (n=173), C- represent patients who received two doses of the vaccine (n=112) and D- represent fully vaccinated patients with three doses of the vaccine (n=138).

We also made corrections in the Figure 1, now Figure 3, regarding Kaplan Meier curve. These adjustments involved the removal of data points corresponding to patients who received only a single dose of the vaccine.

Kaplan Meier curve survival analysis according to the number of received vaccine doses whereas unvaccinated vs. 3 doses p=0.001, 2 doses vs. 3 doses p=0.009.

  1. “There are no concrete references on the use of antiviral and timing of administration.”

Response: Thank you for your observation. In the Discussion we incorporated a statement regarding the utilization of antiviral drugs for the patients in our study: ” Upon admission to the hospital, only patients with the most severe conditions were pre-scribed antiviral medication based on its availability, given the shortage of medications for COVID-19 during that period.”

  1. “Regarding the thrombosis-haemorrhage issue, there is no indication of the type of anticoagulant used in dialysis nor whether there were cases of HIT in the patients.”

Response: Thank you for your observation: We added two more sentences in the paragraph explaining the usage of anticoagulants during hemodialysis: ”In the absence of prior indications of bleeding, heparin was administered to all patients during hemodialysis. On non-dialysis days, low molecular weight heparin (LMWH) was prescribed, with continuous monitoring of anti-Xa levels to ensure they remained within the prophylactic range. No cases of Heparin-Induced Thrombocytopenia (HIT) were recorded’’

  1. “Some acronyms are not adequately explained in the text.”

Response: Thank you for your observation. We made a correction regarding acronyms and explained them adequately.

Reviewer 2 Report

Comments and Suggestions for Authors

The authors presented an interesting study on the factors influencing mortality in the course of COVID-19 among a large group of hemodialysis patients in the era of availability of vaccines against COVID-19. The study is well planned and is characterized by a well-conducted statistical analysis. The novelty of this study may be questionable. The description of the study and discussion also require revision:

1. The novelty of the study is questionable. There are many studies on similar topics. The authors should indicate what the study brings new to the current state of knowledge.

2. The authors conducted many analyses. However, they should indicate what is the primary goal (main hypothesis) of the study and what is secondary analyses. Following this path, authors should indicate what the main and secondary results of their study is. In this way, the discussion should also be revised to refer mainly to the hard endpoint i.e. predictors of mortality... and in addition to secondary analyzes regarding predictors of pneumonia or other complications.

3. The authors should indicate how previous asymptomatic COVID-19 infections were excluded in the study group. The immunity induced by such infection could have a significant impact on the analyzed parameters and patients prognosis. 

4. The authors should indicate which variants of Covid-19 infection dominated in their country during the study period. Hasn't this changed over time? Did it have no impact on the patients' prognosis and should not be included in the analysis or discussion?

5. It is not entirely known whether the analysis applies to all patients with COVID-19 infection or only to those requiring hospitalization (in other words: whether all patients with COVID-19 infection were hospitalized, including those asymptomatic or in good clinical condition).

6. In the discussion, the authors should pay more attention to the high mortality rate due to COVID-19 in hemodialysis patients - as I understand it, the main end point of their study. It is known from previous studies that the fatality rate in the period before the introduction of vaccinations was as high as 30 % and among the oldest patients, even exceeding 40%: DOI: 10.20452/pamw.16028. In the presented study, fatality rate is still high and amounts to approximately 26%, even though a large proportion of patients were vaccinated. Is this due to the fact that the analysis concerns only patients requiring hospitalization?

7. When discussing the results of previous studies on mortality predictors, it is necessary to distinguish between the period when these studies were performed (the period before and after the introduction of vaccinations) and the virus variant that dominated in the studied period. This could have influenced the results obtained.

8. The authors did not refer to an important prognostic factor, which is the frailty index. Its prognostic value has been proven in several studies eg.: DOI: 10.3390/jcm11020285  including the large European ERACODA study: DOI: 10.1093/ndt/gfaa261. If it was not analyzed in the study, it should at least be brought up in the discussion.

Author Response

Dear Sirs/Madam,

Thank you for your kind suggestions to improve our manuscript. Please find attached the document containing all the corrections that we have made.

Reviewer 2

The authors presented an interesting study on the factors influencing mortality in the course of COVID-19 among a large group of hemodialysis patients in the era of availability of vaccines against COVID-19. The study is well planned and is characterized by a well-conducted statistical analysis. The novelty of this study may be questionable. The description of the study and discussion also require revision:

  1. “The novelty of the study is questionable. There are many studies on similar topics. The authors should indicate what the study brings new to the current state of knowledge.”

Response: Thank you for your comments. The prevailing focus of existing research has predominantly centered on examining and analyzing the immune response following vaccination and the subsequent development of antibody titers. In contrast, a more limited body of studies has delved into the examination of crucial clinical events, particularly those culminating in fatal outcomes. Our study aims to bridge this gap by undertaking a comprehensive investigation. We aspire to discern the optimal number of vaccine doses requisite for effectively reducing the risk of fatal outcomes from COVID-19 within a specific demographic of vulnerable patients. Furthermore, our research extends to the identification of key laboratory parameters with the potential to serve as predictive indicators for lethal outcomes, including an assessment of their sensitivity and specificity in anticipating such severe consequences.

  1. “The authors conducted many analyses. However, they should indicate what is the primary goal (main hypothesis) of the study and what is secondary analyses. Following this path, authors should indicate what the main and secondary results of their study is. In this way, the discussion should also be revised to refer mainly to the hard endpoint i.e. predictors of mortality... and in addition to secondary analyzes regarding predictors of pneumonia or other complications.”

Response: Thank you for your suggestions. We incorporated in the text: ”The main goal of the research is to ascertain the optimal number of vaccine doses for dialysis patients that offer enhanced protection against lethal outcomes. Furthermore, the study aims to identify independent predictors of lethal outcomes in these patients, adjusted for vaccination status. Secondary objectives encompass evaluating the impact of the quantity of administered SARS-CoV-2 vaccine doses on averting the emergence of severe clinical manifestations of COVID-19 infection, as evidenced by the severity of respiratory failure and adverse vascular events.”

In the Discussion, we structured the paragraphs to highlight the predictors of mortality we identified and conducted secondary analyses focusing on predictors of pneumonia or other complications.   

  1. “The authors should indicate how previous asymptomatic COVID-19 infections were excluded in the study group. The immunity induced by such infection could have a significant impact on the analyzed parameters and patients prognosis.”

Response: We agree with your comment and appreciate suggestion All hemodialysis patients underwent systematic screening for specific epidemiological and anamnestic data related to potential exposure or symptoms of COVID-19 during each visit. If there was suspicion of exposure or infection, immediate testing for SARS-CoV-2 was carried out. As an added precaution, a weekly screening for SARS-CoV viruses was also implemented. Therefore, we believe that the number of previously infected patients that went undetected was minimal.

We added in the text Study protocol: ”By implementing this strategy from the very beginning of the epidemic in Serbia, the number of potential undetected asymptomatic individuals infected with SARS-CoV-2 was minimal.’’

  1. The authors should indicate which variants of Covid-19 infection dominated in their country during the study period. Hasn't this changed over time? Did it have no impact on the patients' prognosis and should not be included in the analysis or discussion?”

Response: You are absolutely right. Over time several variants of Covid-19 infection were registered.   During period of the study, according to global epidemiological indicators, the beta and delta variants of the SARS-CoV-2 virus were predominant. Unfortunately, during that period, there was no possibility of precisely determining which variant of the SARS-CoV-2 virus each individual patient was infected with.

  1. “It is not entirely known whether the analysis applies to all patients with COVID-19 infection or only to those requiring hospitalization (in other words: whether all patients with COVID-19 infection were hospitalized, including those asymptomatic or in good clinical condition).”

Response: Thank you for your comment. Regardless of symptoms, dialysis patients who tested positive for SARS-CoV-2 were directed to the COVID hospital in Batajnica for isolation from other patients undergoing dialysis. We added in the text this clarification: “Patients undergoing chronic hemodialysis treatment, in the case of a confirmed infection with the SARS-CoV-2 virus, regardless of whether they have symptoms or not, are referred to the COVID hospital in Batajnica for further treatment.”

  1. “In the discussion, the authors should pay more attention to the high mortality rate due to COVID-19 in hemodialysis patients - as I understand it, the main end point of their study. It is known from previous studies that the fatality rate in the period before the introduction of vaccinations was as high as 30 % and among the oldest patients, even exceeding 40%: DOI: 10.20452/pamw.16028. In the presented study, fatality rate is still high and amounts to approximately 26%, even though a large proportion of patients were vaccinated. Is this due to the fact that the analysis concerns only patients requiring hospitalization?”

Response: Thank you for this observation. And it is true, unfortunately, that in our cohort of subjects, a persistently high mortality rate was observed, consistent with the literature. However, among patients who received three doses of the vaccine, this percentage was significantly lower, in accordance with other researchers

  1. “When discussing the results of previous studies on mortality predictors, it is necessary to distinguish between the period when these studies were performed (the period before and after the introduction of vaccinations) and the virus variant that dominated in the studied period. This could have influenced the results obtained.”

Response: Thank you for this observation, which is very important. In general, we have been comparing our results with studies that covered periods dominated by pre-Omicron variants of the SARS-CoV-2 virus.

  1. “The authors did not refer to an important prognostic factor, which is the frailty index. Its prognostic value has been proven in several studies eg.: DOI: 10.3390/jcm11020285 including the large European ERACODA study: DOI: 10.1093/ndt/gfaa261. If it was not analyzed in the study, it should at least be brought up in the discussion.”

Response: Thank you for your comment. Unfortunately, we did not use the frailty index as a significant predictor of clinical outcomes. In the discussion section, as you suggested, we analyzed the findings of two highly relevant studies focusing on these topics: “Multiple comorbidities and the duration of chronic kidney disease (CKD) until the terminal stage is a risk factor for the development of diseases in other organs and organ systems, primarily the cardiovascular system. These patients are frequently in a compromised overall state. In this context, the frailty clinical index is a significant comprehensive indicator of the general condition of dialysis patients, and several studies have demonstrated its significance as a predictor of a lethal outcome.”

Reviewer 3 Report

Comments and Suggestions for Authors

This study focuses on 442 chronic hemodialysis patients with COVID-19, highlighting the impact of vaccination on clinical outcomes. Full vaccination (two or three doses) is associated with improved results, including a reduced incidence of pneumonia and lower risks of complications. Laboratory analyses reveal significant differences between vaccinated and unvaccinated patients in markers such as C-reactive protein, ferritin, and white blood cell counts. Factors influencing mortality encompass comorbidities, pneumonia development, and inflammatory markers. The study concludes that among hemodialysis patients with COVID-19, at least three vaccine doses are protective. Independent predictors of mortality include CRP levels, cardiovascular comorbidities, and lymphocytopenia during infection. While the study provides intriguing data, I recommend further revisions to refine the manuscript:

  1. There have been several studies exploring the effects of COVID-19 vaccination on the clinical outcomes of hemodialysis patients. How does this study differ from other published works? (DOI: 10.1093/ndt/gfab179; DOI: 10.2215/CJN.16621221)

  2. Please specify the inclusion and exclusion criteria for participant selection.

  3. In the methods section, it would be helpful to include a figure detailing the study workflow.

  4. Would it be beneficial to stratify analyzed data based on the type of vaccine received by hemodialysis patients (e.g., mRNA vaccine vs. vector-based vaccine)?

  5. In addition to reporting p-values, please calculate the odds ratio (OR).

  6. Does the term "third dose" in the table refer to the booster dose? If so, consider labeling it as a booster dose to avoid confusion with the primary doses (at least the first two doses for mRNA vaccines).

  7. What confounding factors were identified in this study, and how were they addressed?

  8. Either in the introduction or discussion, please emphasize that previous studies have reported post-vaccination antibody titers that were comparatively lower among hemodialysis patients than in the healthy population. Thus, robust vaccination strategies are needed to address waning immune responses in vulnerable populations (DOI: 10.3390/vaccines11040724).

Author Response

Dear Sirs/Madam,

Thank you for your kind suggestions to improve our manuscript. Please find attached the document containing all the corrections that we have made.

Reviewer 3.

  1. “There have been several studies exploring the effects of COVID-19 vaccination on the clinical outcomes of hemodialysis patients. How does this study differ from other published works? (DOI: 10.1093/ndt/gfab179; DOI: 10.2215/CJN.16621221)”

Response: Thank you for this question. The emphasis of their work was to demonstrate that vaccination is associated with a substantially lower risk of severe COVID-19 outcomes in patients undergoing hemodialysis with SARS-CoV-2 infection. However, the patients in the mentioned studies had received two doses of the vaccine. Additionally, laboratory blood parameters that could indicate the likelihood of a lethal outcome were not considered. Furthermore, in those studies, many patients were not hospitalized, and thus, they did not have the opportunity for daily clinical and laboratory monitoring. In our study, all patients were under continuous medical observation with regular monitoring of blood counts and biochemical parameters throughout the COVID-19 infection. This was due to health policy requirements, necessitating their isolation in a COVID hospital. Additionally, the examined cohort included patients who had been vaccinated with three doses of the vaccine against SARS-CoV-2. This circumstance allowed for the analysis of predictors of fatal outcomes adjusted for vaccination status.

  1. “Please specify the inclusion and exclusion criteria for participant selection.”

Response: Thank you for your comment. We added an explanation: “The inclusion criteria encompassed patients aged 18 and older who had been on a chronic hemodialysis treatment program for more than 3 months and had a confirmed infection with the SARS-CoV-2 virus. The exclusion criteria comprised patients younger than 18 years old and patients who were transiently dialyzed due to acute kidney injury.”

  1. “In the methods section, it would be helpful to include a figure detailing the study workflow.”

Response: Thank you for your suggestion. We added a new figure in Material and Method section with adequate figure caption.

  1. “Would it be beneficial to stratify analyzed data based on the type of vaccine received by hemodialysis patients (e.g., mRNA vaccine vs. vector-based vaccine)?”

Response: We insert one more figure in the Result with the references to the type of the vaccines and their distribution with explanation: “The majority of patients (74%) were administered the inactivated vaccine (Sinopharm), with the remaining individuals receiving mRNA vaccine (Pfizer) at 8.6%, the vector vaccine (Sputnik V) at 5.2%, and vector vaccine (AstraZeneca) at 5.2%. Sinopharm was selected for the first two vaccine doses in 7.1% of patients, with Pfizer being chosen for the third dose, Figure 1.”

  1. “In addition to reporting p-values, please calculate the odds ratio (OR).”

Response: Thank you for your comment. Regrettably, the calculation of the odds ratio was not included in the statistical analysis we conducted, which contrasts with the primary aim of our study. Therefore, we would appreciate clarification on the specific segment of the analysis to which your comment refers. 

  1. “Does the term "third dose" in the table refer to the booster dose? If so, consider labeling it as a booster dose to avoid confusion with the primary doses (at least the first two doses for mRNA vaccines).”

Response: We refrained from categorizing the third dose as a booster due to a lack of comprehensive understanding regarding the variation in clinical efficacy among dialysis patients, contingent upon the number of vaccine doses administered. Our decision was influenced by the absence of sufficient information to definitively characterize the additional dose as a booster, reflecting our commitment to a cautious and evidence-based approach in light of the unique considerations posed by the dialysis patient population.

  1. “What confounding factors were identified in this study, and how were they addressed?”

Response: Thank you for comment. We examined potential confounders such as age, comorbidities, underlying kidney disease, and laboratory analyses using a Cox regression model, as highlighted in the table 5 and table 6.

  1. “Either in the introduction or discussion, please emphasize that previous studies have reported post-vaccination antibody titers that were comparatively lower among hemodialysis patients than in the healthy population. Thus, robust vaccination strategies are needed to address waning immune responses in vulnerable populations (DOI: 10.3390/vaccines11040724).’’

Response: Thank you for your suggestion. We added in the Discussion the literature data for post-vaccination antibody titers implicating that the values of antibody titers were lower among hemodialysis patients than in the healthy population:” In a comprehensive meta-analysis, Notarte et al. showed that patients on maintenance hemodialysis had lower antibody levels throughout all post-vaccination periods compared to the general population. Among the groups examined, those with chronic kidney disease demonstrated the highest antibody immune response, followed by hemodialysis participant, and kidney transplant recipients had the lowest response post-SARS-CoV-2 vaccination.” Also, in conclusion we added sentence:” Our findings suggest that hemodialysis patients constitute a particularly vulnerable group, necessitating a significantly more robust vaccination approach to prevent severe clinical events, including fatal outcomes.”

Round 2

Reviewer 2 Report

Comments and Suggestions for Authors

the authors' answers and the corrections they made are satisfactory

Reviewer 3 Report

Comments and Suggestions for Authors

The paper has been extensively revised and can now be accepted for publication.